# The Young Carers’ Journey: A Systematic Review and Meta Ethnography

**DOI:** 10.3390/ijerph19105826

**Published:** 2022-05-10

**Authors:** Marianne Saragosa, Melissa Frew, Shoshana Hahn-Goldberg, Ani Orchanian-Cheff, Howard Abrams, Karen Okrainec

**Affiliations:** 1OpenLab, University Health Network, Toronto, ON M4P 1E4, Canada; melissa.frew@uhn.ca (M.F.); shoshana.hahn-goldberg@uhn.ca (S.H.-G.); howard.abrams@uhn.ca (H.A.); karen.okrainec@uhn.ca (K.O.); 2Institute of Health Policy, Management and Evaluation, Dalla Lana School of Public Health, Toronto, ON M5T 3M7, Canada; 3Health Sciences Library, University Health Network, Toronto, ON M5G 2C4, Canada; ani.orchanian-cheff@uhn.ca; 4The Institute for Education Research, University Health Network, Toronto, ON M4P 1E4, Canada; 5General Internal Medicine, University Health Network, Toronto, ON M4P 1E4, Canada; 6Department of Medicine, University of Toronto, Toronto, ON M5S 1A1, Canada

**Keywords:** caregiving, young carer, youth, qualitative, meta-ethnography

## Abstract

Despite growing international interest, the caregiving body of literature lacks a recent understanding of young carers’ experiences and their contact with the health care system. We conducted a systematic review of qualitative studies to (1) synthesize more recent qualitative evidence on young carers’ experience, and (2) to identify how these young carers interact with the health care system in their caregiving role. Using a meta-ethnographic synthesis, a total of 28 empirical studies met inclusion. Key findings helped inform an overarching framework of the experience of young carers as illustrated by a journey map. The journey map is a visual depiction of the stages these young carers go through when in a caregiving role framed by three themes: (1) encountering caregiving; (2) being a young caregiver, and (3) moving beyond caregiving. The caregiving experience is perceived by young people as challenging and complex, which could be improved with more informational navigation and emotional support. Understanding these experiences provides insight into gaps in health services and potential solutions that align with the stages outlined in the journey map.

## 1. Introduction

Worldwide prevalence rates of young carers have been estimated to be between 2 to 8 per cent and rising [1]. Young carers are providing an increasing level of unpaid support to family members and yet they have remained overlooked in both research and policy [2]. In a recently updated 2015 scoping review on young carers [3] found since then, little attention has been paid to why young people are taking on caring roles, and limited data is available on the science of caregiving in children and youth [2]. Instead, public and political interest in and support of adult family caregivers, or “essential care partners” has grown substantially [4]. The term ‘young carer’ refers to those children and young adults under the age of 25 who provide substantial, consistent or significant unpaid assistance to a family member living with a chronic illness or disability, mental health condition, issues with substance use or those resulting from frailty due to aging [5,6]. Given that the definition for young carers was conceived of in the early 1990s [7,8], its expansion to additional groups and a boarder age limit is a reflection of more recent sociodemographic changes. One jurisdiction surveyed showed that 84% of offspring aged 16 to 29 were still living with their parents [7]. With delayed conception in later years to parents who may need caregiving help, a shrinking pool of potential carers as more women enter the workforce, a growing population of people living with chronic disease, and the financial strain of maintaining a household, unpaid caregiving among youth is expected to increase [6].

For young carers, having a choice and the time to develop caregiving skills rarely happens; instead, they take on “exceptional” duties to keep their families intact and functioning [9]. Young carers are often at an important physical, emotional, and mental developmental stage where life plans are not yet set [10]. Taking on caregiving responsibilities can interfere with life pursuits—postponed education and career development as well as progression in personal relationships [11,12]. Young carers often show less school engagement than their non-carer peers and report higher levels of mental health problems, insomnia, somatic symptoms, and lower life satisfaction [13,14]. Feeling ‘different’ and misunderstood by their peers, withdrawing from their circle of friends, and concealing their caregiving identity have all been reported by young carers in the literature [9,15]. Caregiver responsibilities can lead to less time to study, missing school, arriving to work late or leaving early, and taking time off from work because of caregiving responsibilities [16]. Young people in caring roles are less likely to be employed, have lower earnings from paid work, and report poorer mental and physical health outcomes compared to young people who were not providing care [17,18].

Despite evidence of poorer health and well-being in the literature, young carers remain a hidden population in health care and policy agendas. This may be due to the lack of timely and broad understanding of young carers’ experiences, including those that may shape their interactions with the health care system such as care providers or services [17]. While young carers’ intersection with the education and social care system has been reported in previous studies [19,20], their contact with health systems is less known even when they have knowledge to contribute or provide support [21]. Research on the awareness of young carers among health care providers is growing, though still in its infancy [22,23]. A meta-ethnography approach seeks to produce higher-level abstracts and understanding while maintaining the richness of primary studies [21]. Therefore, a meta-ethnography was selected as an appropriate method to draw on the words of young carers to convey their collective experiences, and to report on their interactions with the health care system.

Given the evolution in the prevalence, role and recognition of young carers over the years [24] and in the last ten years since the research was synthesized [25], the purpose of this meta-ethnography was to (1) synthesize more recent qualitative evidence on young carers’ perceived experience of caring for a family member with a chronic illness or disability and (2) to identify if and how these young carers interact with the health care system in their role as caregiver. This review of the literature will contribute to our understanding of young carers’ involvement with formal health services in their role as caregiver. Our overarching goal was to use young carers’ experiences to help inform points of potential intervention in health care and policy.

## 2. Materials and Methods

### 2.1. Meta-Ethnography

A meta-ethnographic approach was used to systematically compare studies that pertain to the study aim [26]. In contrast to other narrative-type literature review methods, meta-ethnography attempts to achieve a new interpretation of the selected studies, moving towards a reconceptualization [27,28]. Noblit and Hare [26] outlined a seven-step process for conducting a meta-ethnography (Box 1). The product of this synthesis is the translation of studies into one another, which requires the researcher to identify common ideas, concepts, and metaphors across studies [27]. 

Two distinct phases were undertaken: the search and the synthesis of findings. Meta-ethnography, as described by Noblit and Hare [26], is used in the process of interpreting and synthesizing multiple qualitative studies. Meta-ethnography is a well-accepted tool for interpreting multiple cases or multiple accounts of studied phenomena [29,30]. 

Box 1Seven steps of Noblit and Hare’s meta-ethnography [26].

Getting startedDeciding what is relevant to the initial interestReading the studiesDetermining how the studies are relatedTranslating the studies into one anotherSynthesizing translationExpressing the synthesis


### 2.2. Literature Search

A comprehensive search strategy was developed to find qualitative studies on young carers. The initial search strategy was developed for Ovid Medline by using a combination of database-specific subject headings and text words. Due to the difficulty of specifying the age of caregivers through database searching, some trade-offs on comprehensiveness, such as tighter adjacencies and not using subject headings for caregivers in isolation, were taken to avoid too many unrelated citations on adult caregivers of children and adolescents. The search strategy for young carers was then combined with a filter for qualitative studies which was modified and expanded [31]. The Medline search strategy was then customized for each database. 

Searches were executed on 18 October 2019 and updated 13 November 2020 in the following databases: Medline (Ovid), EMBASE (Ovid), PsycINFO (Ovid), CINAHL (EBSCO), Emcare (Ovid), Scopus, and Web of Science. Limits were applied for English language, years 2008–present. We purposefully used 2008 as a follow-up to a previously published seminal systematic review that was conducted using no defined limits during December 2008 [26]. Conference and book materials were excluded where possible. See Appendix B for search strategies.

### 2.3. Selection Criteria

Our electronic database search yielded a total of 4116 citations. Two authors (M.S. and K.O.) initially screened together 25% (*n* = 1008) of results at the title and abstract stage to ensure the presence of internal reliability. Studies at the full-text review stage were screened independently by two reviewers against inclusion criteria. Twenty-eight papers representing 25 studies met eligibility. 

The following inclusion criteria was applied: (1) application of qualitative research methods; (2) explores the first-person account of being a young carer; (3) care recipient is a relative with an acute or chronic illness; (4) written in English, and (5) has been published in a peer-reviewed journal and have ethical approval from an appropriate agency. 

Studies using mixed methods were accepted, however, only the qualitative findings were considered. The following were the number of full text articles that were excluded due to no meeting study inclusion due to study sample (*n* = 30), study design (*n* = 24), experience (e.g., not focused on the experience of being a young carer) (*n* = 22), publication type (*n* = 13), language (*n* = 1), and already included in the review (*n* = 1). See Appendix A for PRISMA flow diagram.

### 2.4. Data Analysis

Two study authors (M.S. and K.O.) used a table created in a Microsoft Word 365 document to extract data from a subset of articles from the final set of studies. Authors finalized the table and collected the followed study characteristics: country of origin; study purpose; study design; methods; sample (description of carers and care recipients). In a secondary matrix form, as described by Britten and Campbell [27], we extracted ‘second-order constructs’ (i.e., authors’ interpretation of primary data) developed from ‘first-order constructs’ (i.e., participants’ interpretation of their experiences). At this stage, we used thematic analysis to develop categories from the first and second-order constructs. These categories initially included changing family roles, worries and concerns, enacting the caregiving role, and being in a state of readiness. 

The next step involved constructing third-level constructs by comparing categories and concepts in one study to those of another to uncover similarities or differences in their portrayal and meanings. During this stage, we continued to refine the categories into themes by merging and revising the categories. Synthesizing the data resulted in a table featuring third-level constructs, or the emergent themes, and the associated second-level constructs. Supporting direct quotes were also included. Since definitions of first-, second-, and third-order constructs vary in the literature, we provide our adapted definitions in Table 1 [32]. The collective experience of young carers throughout the caregiving continuum were used to generate a young carer journey map to depict the interpretation visually as a means of building on the previous knowledge base of what is known of young carers [25], and making the findings and associated support needs more accessible to the reader.

### 2.5. Overview of Synthesized Studies

After applying the inclusion criteria, a total of 28 studies remained. Publication date ranged from 2009 to 2020 with most studies originating from the United Kingdom (UK) (*n* = 9) [33,34,35,36,37,38,39,40,41], followed by the United States of America (4) [42,43,44,45], Finland (*n* = 3) [46,47,48], Canada (*n* = 3) [45,49,50], and Sweden (*n* = 3) [45,49,50,51,52,53]. The remaining studies were published in either Iceland [54], Israel [55], Turkey [56], China [57], New Zealand [58], Kenya [59], or Australia [60]. However, three of the studies from Finland [46,47,48] and two from Sweden [51,52] resulted from the same datasets. Studies reported a range of study designs including ethnography [34,46,47,48,57,59], grounded theory [33,58], phenomenology [54], qualitative research [36,37,40,44,49,55], case study [39,56], mixed methods [38,42,45,51,52,60], and a narrative methodological or analytical approach [50,53]. In total, this meta-ethnography has worked to synthesize the voices of approximately 1166 current or previous young carers who provided care to physically and mentally ill community members and parents living with a variety of conditions including AIDS/HIV, young onset dementia, multiple sclerosis, mental illness, substance misuse, and Huntington’s disease (see Table 2 for study characteristics).

## 3. Results

The synthesis revealed three major themes with associated sub-themes: (1) Encountering the disease; (2) Being a young carer, and (3) Moving beyond caregiving. We depicted these themes along with our interpretation of associated needs in a journey map, which consists of the sub-themes and illustrative quotes. See Appendix A for Young Carers’ journey map figure. 

### 3.1. Encountering Caregiving

#### 3.1.1. Accepting the Role of Carer

For many young carers, the first step in the ‘encountering the disease’ phase included accepting the role. Varying motives were noted across studies that mostly reflected either filial obligation [33,34,50,54,55] or religious reasons [34,46]. For many young carers, they accepted the role without question and they would “just do it” in an automatic way [55,58]. One young carer described their experience of becoming a caregiver in the following quote, “… stopping my life for my father—I jumped in at the deep end automatically, out of choice” [55]. The extent to which young carers perceived having control over their caregiving role seemed integral to their own sense of “agency” or control according to Blake-Holmes [36]. Similarly, having no choice is also reported by young carers because of the sick parent’s refusal of external help noted in the following narrative: “Our father won’t accept that [outside help] because he’ll say: ‘Why? I have three daughters’… So, we managed everything on our own. We don’t really have the choice’” [50]. Conversely, some respondents strongly identified becoming a carer based on religious values and conceptualization of the caregiver role [34,46].

#### 3.1.2. Being Uninformed and Desiring Support

Many studies reported on young carers with varying degrees of being uninformed or feeling unprepared about the illness and as a result, either not knowing what to expect or only becoming more aware following a parent’s death [33,34,37,39,42,43,49,54,55]. Despite managing through unrewarding care activities for parents with cognitive impairment, carers described feeling less than equipped to do so, “No one prepared me for these mental changes… I lost my mother years ago… and now we have to care for her like a child” [54]. For others, despite actively caregiving, they had little to no knowledge of the current situation: “… we didn’t know that there was, um, that it was life-limiting or anything like that at first… until he died” [37].

Desiring more support was also reported in several more recent studies [33,39,42,43,49,50]. Young carer participants expressed interest in either informational, a booklet for example, or emotional sources of support. In the work of Lally and Hydeman [43], participants mentioned wanting to either read about the cancer diagnosis or to talk to someone about what they were experiencing. One participant identified the lack of available support for black and minority ethnic young carers: 

“[They say], ‘Let’s not talk about it. Let’s pray. Let’s think positive.’ There’s nowhere to go in the community to speak about things to anyone. It’s like no one knows that African Americans and minorities are having cancer. You don’t know where to go” [43].

For those who described needing to share their worries, connecting with someone with a similar experience seemed important. The following quote underscores, first, young carers addressing gaps in knowledge, and taking on the added responsibility of explaining the patient’s behavior to family members: “I know that he has like mood swings and things like that, side effects of his medication as well, that are the things I try and find out and try and explain them to my mum. She does understand them but it is hard for her to keep them in mind obviously every day is difficult” [33]. Very few participants acknowledged accessing support. Instead young carer participants reported that services were targeted to the care recipient rather than the carer [39].

#### 3.1.3. Adjusting to a New Reality

The adjustment to a new reality manifested in different ways, for example, with changes to the ill family member and day-to-day life, as well as the carer as an individual. Several young carers noticed ‘Changes in behavior’ in chronically ill parents, which prompted some degree of self-reflection [33,45,54,57]. Young carers dealing with early onset dementia noticed slowly emerging changes often dismissed or attributed to stress and fatigue [33]. For those caring for a family member with Huntington’s disease, they expressed realistic concerns about developing this illness and experiencing similar limitations, challenging behaviors and dependencies [45]. Changes in the family member had an impact on how the carer related to that individual, as well as family dynamics and relationships among other household members: “The loss of my dad’s personality, his essence is gone but also the loss of that particular family dynamic and the relationship we had. Completely unconnected to his physical presence. There’s no emotional connection there” [39].

Several studies also found that young carers had to adjust to a new day-to-day reality, which meant more responsibilities, forcing them to grow up—becoming the “man of the family” or “not really having a chance to be a kid” as described by Allen and Oyebode [33]. For some, they willingly accepted taking on more tasks [40], others reported resentment over ‘imposed maturity’, “She just… she can’t look after herself, she can’t look after the house, she can’t cook or anything… I think I’m just upset that I have to take on more and more responsibility as mum gets worse” [60].

Young carers also described changes at a personal level when encountering this new reality. One young carer, for example, noted that she had to instill her own sense of stability by establishing rules her own friends were expected to follow: “I had to parent myself… usually did what my girlfriends had to do… go home at ten…not watch too much television” [54]. These adjustments also included accepting expectations of a new normal for the young carers while also still wanting to be cared for by the ill parent. In one example by Newman and Bookey-Bassett [49], a chronic illness causes a young person to want their mother suffering from dementia to take care of them: “I have [a chronic illness] so I am in pain, she’ll come and like give me a hug and she’s like it’s okay and she asks what she can do. So, she’s still like for us to be my mom and I am still like want her to have that too, to be able to take care of me. It’s just different now”.

### 3.2. Being a Young Carer

#### 3.2.1. Performing Daily Tasks and Duties

Narrative on carrying out practical caregiving tasks dominated the findings across most studies. These duties could be categorized as medical in nature, personal care, emotional support, household chores, caring for siblings, and to a lesser extent, navigating the health care system. Medication administration and managing problematic behavior formed the medical care tasks [33,39,40,43,44,48,55,59]. The findings described in two studies support the importance of medication when caring for family members living with HIV/AIDS [48,59]. Often medication care actions were considered in relation to nutritional needs whereby young carers made an effort to ensure patients take the antiretroviral medication on a full stomach [48].

Providing intimate personal care also was described by young carer participants in the following quotes: helping her to the toilet,” [54], “I have to feed her and watch her for my dad to work” [45], and “… care means washing their body, hair, and clothes. It is also helping them to dress sometimes” [47]. Children as early as preschool age reported how they cared for their alcohol-dependent parent, getting them undressed or cleaning up vomit [53]. For those caring for family members living with HIV/AIDS, lack of personal protective supplies instills fear and impacts their quality of care, “I don’t have gloves. But, she has an oozing wound and also diarrhea. So I have fear that I may be infected when washing and changing her clothes. Because I am afraid of the infection I have not changed her clothes for three days” [48]. 

Providing psychological comfort and support also formed a major caregiving task for young people. Despite their youth, many participants seemed able and willing to emotionally connect to either the ill family member or to others around them. For example, young carers tried to offer support to the non-affected parent [39]. When caring for persons with AIDS/HIV, praying and attending religious establishments was also cited as a form of emotional support [47,48]. 

Taking on a parenting role was also part of daily duties noted in quotes that speak to caring for younger siblings [37,52,53,54,56,57]. In these examples, young carers describe being an “extra mom” [52] and “speaking to teachers” [37]. In the study by Allen and Oyebode [39], the concept of ‘parentification’ is described in relation to the young person taking responsibility for both parents’ health. Advocating for mental health care for their non-affected parents was the example described in the following quotes, “And later it was me that actually that got her seeing some counselors because I actually stood up to Dr X an’ [sic] said ‘Well what are we meant to do then?’” [33]. 

Very few participants reported specific interactions with the health care system in their caring role; however, for those that did, the data seem to reflect young carers’ role either during medical appointments or interactions with community providers when learning to administer medication [44,49]: “Because like I said, my mom can’t do stuff by herself. So she has to have aides in the day. And there’s this one aide… She’s been around since like, since the beginning and she’s still with us. So I learned from her” [44].

#### 3.2.2. Experiencing the Personal Toll

The personal impact of being a young carer seemed to manifest in participants having worries and concerns, having to sacrifice themselves, and experiencing social and economic repercussions. Most of the studies reported on young carers’ emotional response to being a caregiver [33,34,37,38,40,42,43,45,47,50,52,54,55,57,59]. For some, they worried about the affected person or other family members, such as the non-affected parent or siblings, having trouble coping and experiencing low mood [33].

It appeared that young carers described their own mental health issues in relation to their caregiving duties. This meant they had trouble concentrating, and for some, were diagnosed with depression, anxiety requiring antidepressant medications [33,38,45,57]. Perhaps compounding their emotional struggles for some carers was either actively trying to hide the illness or feeling embarrassed by their situation. In dealing with their parent’s severe mental illness, child participants expressed suppressing their own emotional needs to avoid people knowing and to maintain their child–parent relationships:

“I understand clearly she is ill. It is not her fault. She does not want to roll around on the floor… However, she is so noisy. I have to go to school, I need sleep… Sometimes, I am very angry with her” [57].

Moreover, young carers described some degree of self-sacrifice, whether in suppressing emotional needs, forgoing peer relationships, or making economic or personal offerings [34,36,37,43,45,46,47,57,59]. Home became a “prison” for some young carers who felt trapped by the ill family member [55], or unable to meet with friends because of having to care for a disabled sibling [56]. Young carers living in extreme poverty were noted to sell property and their bodies for income, and even described leaving school prematurely because of an unwanted marriage. In the latter case, marriage was seen to secure help from a partner for a sick parent [59]. Economic and social repercussions of being a young carer emerged in many studies across different contexts—that is, in families dealing with HIV/AIDS, young onset dementia, and Huntington’s disease. Limited access to school, food, and financial strains were concerns mentioned, which forced some young carers to rely on external relatives or friends for support [38].

#### 3.2.3. Identifying Positive Gains

Some studies found that caregiving could be a rewarding experience for young carers as they identified a number of positive gains [37,39,40,41,45,47,49,52,54,55,57,58,60]. For some participants, caring was an opportunity to receive satisfaction, while many others spoke of having a stronger bond with the ill family member, “We have this unbreakable link and I try to visit him every day… as his safety is my security” [35,54]. 

Quite striking is the insight many young carers had about their own personal growth [36,37,45,52,57,60]. As many young carers grapple with the responsibilities, they were able to describe ways in which the experience changed their mindset and outlook on life. For example, several participants identified these constructive changes, “A lot of people say that empathy is one of my biggest qualities and that I am very empathetic towards other people” [37]. Similarly, the work of Smyth, Cass [60] reported that young carers engage in ‘self-appraisal’ to assess their level of responsibility in comparison to their peers. Finally, one group of participants described the personal transformation brought on by caring for a parent dying of cancer was a formative experience [55].

By way of contrast, not all young carers engage in benefit-finding as a carer. For example, for one participant from the same study, the caregiving experience had a very different meaning, “How can treating a dying parent be perceived as a positive experience? It is impossible; terrible. I feel that this turned me into a less positive person” [55]. For some, they struggled to identify unexpected rewards from the experience. 

### 3.3. Moving beyond Caregiving

#### 3.3.1. Planning for the Future

A subset of studies reported on young carers considering and actively planning for their future in the context of being a carer [33,36,37,38,39,45,50,55]. In these passages, young carers express their personal desires for doing well in college, having their own families, and being relieved of the burden of care. However, they often seemed torn between relief and guilt as their lives seemed to continue past caregiving, “Last time my mum was ill, I was like I need to go out there and he was just like, just don’t… I feel guilt all the time with my mum, and I don’t know why because I’ve got no reason to be beholden to her, and I don’t owe her anything” [36]. For others, as they spoke about what the future held for them, they continued to feel obligated to support their families and used education as a mechanism [37,38,45,54]. According to the study by Clabburn and O’Brien [37] and Goldblatt and Granot [55], being a young carer had direct implications on their education such that several participants decided to pursue medicine and study palliative care. 

#### 3.3.2. Reflecting on the Past

Seven papers described young carers looking back on their caregiving experience [33,36,49,50,51,52,55]. Narrative centered on factors that could have changed or improved the caregiving situation, such as having more support for their ill parent and having their parent still alive, “He would have taught me things. Taught me the business, I could have asked him about things—girlfriends. Adult male talk. Working with dad” [33]. In these passages, young carers reflected on ‘what could have been’ if they were not dealing with an ill family member, “So, like in the future, or like would my mom help me with like these things like moving into my university/house like things like that, that she didn’t help with so I think that like that’s where it hurts the most” [49]. Missed opportunities were also mentioned by participants in terms of having to grow up quickly, or of not pursuing education or personal relationships. According to study authors Blake-Holmes [36], caregiving affected young carers’ ‘exercising agency’, a process of social engagement with the world around them. Their sense of relentless demands hindered participants’ transition into adulthood by allowing little time for their personal agency.

## 4. Discussion

The aim of this meta-ethnography was to examine the qualitative evidence on young carers’ experience of caring for a family member with a chronic illness or disability. This synthesis helped inform an overarching framework of the experience of young carers as illustrated by a journey map (see Figure 1). The journey map is an illustration of the steps these young carers go through when in a caregiving role, framed by three overarching themes: (1) encountering caregiving; (2) being a young caregiver, and (3) moving beyond caregiving. In mapping the experience over time using the lens of health care, the team was able to interpret the themes to identify needs of young carers along the journey, and suggest points of intervention and opportunities to address challenges faced by young carers. 

The ‘encountering caregiving’ theme is supported by the following three sub-themes: Accepting the role of carer, being uninformed and desiring support, and adjusting to a new reality. Our synthesis revealed that young carers often automatically accepted the caregiving role without much questioning or objection. We noted that participants’ minimization of their caring role and additional responsibilities is reflective of both their own lack of self identification and their invisibility, more broadly. These findings are echoed in previous research that reports that caring, for young people, is integrated within a normative framework of familial obligation and responsibilities [61,62]. Young carers diminish their work by viewing caring as an “extension” of their role within their family; rather, recognizing one’s duties as being “caring” as rarely spontaneous [61,62]. In addition, our findings indicate an absence of young carers’ interaction with the health system despite expressing a strong need for both informational (information on how to prepare for caregiving and education to improve health literacy) and emotional (validation of their caregiving role and coping mechanisms) support. 

Adjusting to a new reality involved “Being a young carer,” which for many young carers represented overseeing domestic chores, household management, and personal care, as well as managing the personal impact of the role, both negative and positive. Our results reinforce the complex nature of caregiving for young people who, in addition to practical duties, includes a high mental and emotional load that cuts across socio-cultural contexts. To date, previous research has found that young carers report poorer physical health, mental health, and psychosocial outcomes [63,64], and have lower educational aspirations and attainment when compared to non-carers [65]. Our team found that the experiences of young caregivers point to specific informational and emotional needs such as the need for peer support, self-care tools, skill building, and caregiver training. This is in agreement with previous research, which has shown that young carers have unique support needs, e.g., connecting with similar peers, accessing counseling, and being acknowledged for their role. Importantly, finding a sense of belonging in the carer identity can be especially validating as they navigate their desire for normality [66]. However, an increasing global prevalence of young carers [67] failing to address their needs will lead to more negative adjustments across a range of emotional and mental health outcomes [63]. Our findings demonstrate a pressing need and opportunity for the health system to provide support services to young carers that can serve a preventative role.

Our third theme, “Moving beyond caregiver,” elucidated the transition period these participants experienced. Some young people were able to create a meaningful future stimulated by the caring role, which is very much aligned with recent research on “moral resilience” among young carers [68]. This is to say that young carers can develop a positive social concept of caring; however, young carers do express tension between their want for a ‘normal’ life and their caring role affecting their emotional well-being [69]. Our findings underscore the need for better provision of formalized care to reduce the need for young people to take on the carer role, and to invest in greater supports for those who do provide care. Our findings also highlight specific needs of young carers during this stage such as counseling and education on life skills and career paths that may help with the transition. Some of the data pointed to either contemplating leaving home or having made the move to more independent living. Previous research also found leaving home was problematic because of the perceived relentless responsibilities by young carers [70]. A combination of dealing with a family member’s illness, lack of support services, and facing critical developmental stages can propel young carers into more vulnerable situations, such as being homeless or underhoused.

### 4.1. Study Strengths and Limitations

To our knowledge, this meta-ethnography represents a more recent synthesis of young carers’ experiences [25]. One strength of this meta-ethnography was the larger number of international papers, for example, from Finland, China, Israel, and Turkey. Despite the apparent heterogeneity of social-cultural features between studies, we were able to identify common themes and experiences. On that note, this synthesis also offered a wider age range of participants allowing for some reflection from former carers. Caring experiences across the continuum are described in the narrative themes and identified along with their support needs in a journey map. Our journey map does not represent what happens to every young carer, but it does provide a guide for where interventions can be targeted to better support young people in their caring role. Our research team involved in the analysis and interpretation was formed by content (young carers), methodological (qualitative research), and health care experts. Although our meta-ethnography was informed by a collaborative approach to interpreting the constructs, authors’ diverse professional and personal backgrounds may have introduced biases. Therefore, we propose that future research could challenge our interpretation and version of a journey map and consider other intersections with a social and education lens [71]. 

Our paper is not without limitations. While we had two reviewers screen title and abstracts to establish internal validity, one reviewer screened the majority of papers. Therefore, relevant studies may have been missed including those studies with a wide participant age range. Next, while we attempted to generate a high-level understanding of young carers’ experiences, important sociocultural nuances may not have been captured. For example, the role of religious beliefs and cultural expectations on young carers’ decisions to become a caregiver may differ based on where the study was conducted. However, it should be noted that, in general, young carers in different countries carry out the same range of caring activities [72]. Many of the included studies identified ‘qualitative’ methods as a study design. This is problematic in its broadness, and we acknowledge that this may have impacted on our synthesis of primary qualitative research. Last, a quality appraisal of the original studies was not conducted, and so, we are unable to comment on the included studies’ methodological quality or appropriateness to their research aims.

### 4.2. Policy and Practice Implications

Findings from our meta-ethnography and resulting depiction of the journey of young carers highlight how despite their invisibility, young carers play an important role in the health care system and have very real needs. It also suggests that we have limited data on the nature of young carers’ involvement with the broader health system and care and service providers. These insights have the potential to provide invaluable information to inform future research and practice. For example, encountering the caring role is an opportunity for these youth to be better prepared and supported by health care professionals. Given that professionals are not always aware of the potential caring roles of young people [73,74], a first step could involve creating practice tools to help professionals recognize young carers and connect them to appropriate support services. For example, adapting existing patient assessments to include a question to identify young people with caring responsibilities [70]. Existing guidelines, for example, by the UK’s The Children’s Society [75], can serve as a template offering information of who are young carers, situations where support for young carers and their families might be needed, and lists of organizations that can provide this assistance.

## 5. Conclusions

Despite the growing number of studies in this population—the role they play and their support needs—young carers largely remain unrecognized by the health system. However, their experience and desire for greater support is heightened by this meta-ethnography, and our findings identified specific needs along the young caregivers’ journey and potential opportunities. 

## Figures and Tables

**Figure 1 ijerph-19-05826-f001:**
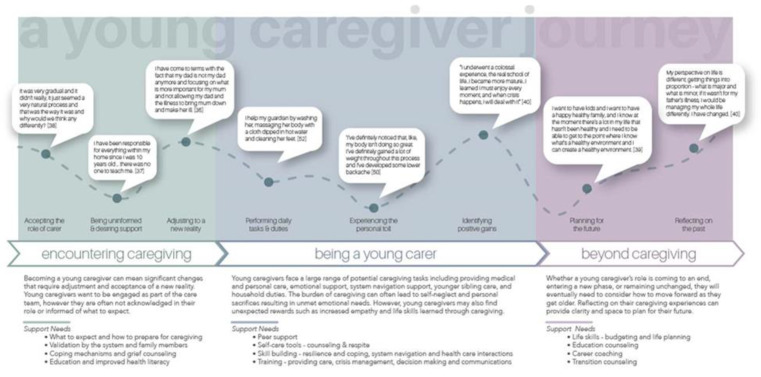
Young carers’ journey map.

**Table 1 ijerph-19-05826-t001:** Meta-Ethnography Definitions.

Term	Definition
First-order constructs	The participant’s interpretation of their experience with caregiving
Second-order constructs	The author’s interpretation of young carers’ experience with being a carer, described as themes and concepts using the original terminology presented in the papers.
Third-order constructs	The synthesis team’s interpretation of how the studies relate by comparing first- and second-order constructs and conveying overarching themes and concepts about a phenomenon.

**Table 2 ijerph-19-05826-t002:** Study characteristics.

Author/s	Year	Country of Publication	Purpose	Study Design	Methods	Sample Carers (N, Gender, Age Range)	Sample Care Recipients
Aga, Nikkonen & Kylma	2014	Finland	To discuss the caregiving actions of the HIV/AIDS family caregivers in Addis Ababa, Ethiopia, as emanating from their conceptions of care and situated within the constraining sociocultural factors in which they live.	Focused/micro ethnographic and symbolic interactionism frameworks	Interviews and participant observation	*n* = 18 Female = 18 16–76 years, with the majority (55.5%) between 16–27 years	Person living with AIDS
Aga, Kylma & Nikkonen	2009	Finland	To explore and describe the cultural and social factors that influence care and caregiving from the perspectives of the family caregivers of PLWA in Addis Ababa, Ethiopia.	Same as above	Same as above	Same as above	Same as above
Aga, Kylma & Nikkonen	2009	Finland	To explore and describe the conceptions of care as a cultural phenomenon from the perspectives of family caregivers of PLWAs in Addis Ababa, the capital city of Ethiopia.	Same as above	Same as above	Same as above	Same as above
Ali et al.	2011	Sweden	To describe the daily life of young people who care for friends or family members with mental illness and explore how they manage in everyday life.	Qualitative descriptive	Interviews, a focus group interview, and telephone interviews	*n* = 12 Female = 9 16–25 years	Persons living with mental illness
Ali et al.	2013	Sweden	To explore how young (16–25-year-old) informal carers of a person with a mental illness experience and use support.	Mixed method approach	Same as above	Same as above	Same as above
Allen et al.	2009	UK	To explore the impact on young people’s wellbeing of having a parent with younger onset dementia.	Grounded theory	Interviews	*n* = 12 Female = 7 13–24 years	Parent with young onset dementia
Andersen et al.	2012	UK	To describes the experiences of children, who in contexts of poverty, play a key role in managing the treatment and care of a sick parent on antiretroviral therapy (ART), as well as how they strive to sustain the household and maintain their own wellbeing.	Anthropological fieldwork	Interviews, focus groups, participant observation and drama	*n* = 20 6–16 years	HIV-infected parents
Barry	2011	UK	To explore the views and experiences of young carers in Scotland from a social capital perspective, in terms of how they experience and negotiate their family, school, and social lives as young people caring for ill or disabled family members.	Qualitative research	Interviews	*n* = 20 Female = 10 12–23	Disabled (mentally or physically) parents
Bjorgvinsdottir & Halldorsdottir	2014	Iceland	To study the personal experience of being a young caregiver of a chronically ill parent diagnosed with MS.	Phenomenology	Interviews	*n* = 11	Parents diagnosed with MS
Blake-Holmes	2019	UK	To understand how adults who cared for a parent with a mental illness make sense of their experience and what impact does this have on them as adults?	Qualitative research	Interviews	*n* = 20 Female = 15 19–54 years	Parents living with mental illness
Chan & Heidi	2010	China	To investigate the needs of a neglected group, namely Chinese adolescent children of parents with schizophrenia, to gain insights into the design of programs for these adolescents	Ethnographical research method	Interviews	*n* = 5 Female = all 12–15 years	Parents living with mental illness
Clabburn & O’Brien	2015	UK	To explore how providing care for a parent with motor neuron disease impacts upon a young person’s life.	Qualitative methodology	Interviews	*n* = 7 Female = 5 20–31 years	Parents living with motor neuron disease
Cluver et al.	2012	UK	To determine educational impacts of household AIDS-sickness and other-sickness/What impacts on their education do adolescents living in sick homes and healthy homes identify?	Qualitative, participant-led methods	Interviews	*n* = 659 Female = 358 10–20 years	AIDS-affected homes
Hagström et al.	2019	Sweden	To investigate what it means to grow up in an alcoholic family environment.	Narrative analysis	2019	Sweden	Alcohol misuse
Gold blatt et al.	2019	Israel	To address the meaning of being young adults, who were the caregivers of their dying parents.	Qualitative	Interviews	*n* = 14 Female = 9 20–35 years	Parents with terminal cancer
Kale & Siğirtmaç	2020	Turkey	To examine the participation process of elder siblings in caregiving for their siblings with special needs and how this participation process reflects in their peer relationships in rural context in Turkey.	Case study	Interviews	*n* = 6 Female = 3	Younger siblings who have disabilities
Kavanaugh et al.	2016	USA	To explore caregiving youth knowledge of end-of-life wishes and their willingness for end-of-life discussions.	Mixed methods	Interviews	*n* = 40 Female = 31 10–20 years	Parent with Huntington’s disease
Lally et al.	2020	USA	To explore the experiences and needs of African American children and adolescents who were identified by a cancer survivor in their family as providing substantial supportive care during diagnosis and treatment.	Not reported	Focus groups, interviews	*n* = 5 Female = 3 7–19 years	Family member living with cancer
Masterson-Algar & Williams	2020	UK	To investigate the impact that having a parent with a neurological condition can have on young adults’ experiences of growing up and the nature of their support networks.	Case study	Interviews	*n* = 31 Female = 20 16–25 years	Family member with a chronic condition
McDonald et al.	2010	New Zealand	To explore the experiences of young carers and those they care for.	Grounded theory	Interviews	*n* = 14 Female = 12 <18–25 years	Family members with a range of medical conditions
McGibbon et al.	2019	UK	To identify the factors that contributed to or challenged the resilience of young carers.	Qualitative study	Interviews	*n* = 22 Female = 16 8–18 years	Family member(s) with physical disability and/or mental illness/ developmental problem
Newman et al.	2019	Canada	To explore the experience of being a young carer of a family member living with dementia.	Qualitative descriptive	Interviews	*n* = 5 Female = All 20–26 years	Family member with dementia
Nickels et al.	2018	USA	To gain insight into the children’s reported experiences and perceptions surrounding their responsibility related to medication management.	Qualitative study	Focus groups	*n* = 28 Female = 20 12–19 years	Multiple diagnoses (e.g., asthma, diabetes)
Olang’o et al.	2010	Kenya	To understand the consequences of children’s caregiving role on their overall growth and development, particularly in those regions worst affected by HIV and AIDS.	Ethnography	Interviews, personal narratives	*n* = 19 Female = 12 10–18 years	Persons living with HIV
Silverman et al.	2020	Canada	To examine the stories of young adult women carers (everyday lived experiences).	Narrative inquiry	Social network maps, participant-driven photography, caregiving timeline	*n* = 10 Female = 9 25–36 years	Variety of illness
Skovdal	2010	UK	To explore children’s contributions to their fostering households.	Not specified	Interviews, photography, and drawings	*n* = 69 11–17 years	Ageing and ailing individuals
Smyth et al.	2011	Australia	To examine the costs incurred and benefits conferred by young people who usually provide care to family members, with disability or chronic illness.	Mixed methods	Interviews, focus groups	*n* = 68 11–25 years	Family members with illness or disability
Williams et al.	2009	USA/Canada	To describe caregiving by teens for family members with Huntington’s disease.	Mixed methods	Focus groups	*n* = 32 14–18 years	Parent or grandparent living with Huntington’s disease

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
