# Peer review of "The Young Carers’ Journey: A Systematic Review and Meta Ethnography"

_ijerph, 2022, doi:10.3390/ijerph19105826_

Round 1
Reviewer 1 Report
dear colleagues,
Qualitative synthesis methods are becoming more popular in research. Meta-ethnography is one of the most often used methods. This is a systematic strategy that combines data from different research to provide new insights into the experiences and viewpoints of participants. the paper was based on a systematic review of qualitative studies to 1) synthesize more recent qualitative evidence on young carers’ experience, and 2) to identify how these young carers interact with the health care system in their caregiving role. The findings imply that young people view caring as a difficult and complex experience that may be enhanced with additional information, navigation, and emotional support. Understanding these experiences might help identify healthcare gaps and potential solutions that correspond to the stages of the trip map.
The title needs to be a systematic review and meta-ethnography.
The introduction did not clearly address why a qualitative methodology is appropriate. Why not quantitative analysis (meta-analysis). The aim is not clear too. Page 2, the last paragraph is very broad. The goal of the study is to analyze or illuminate research participants' activities and/or subjective experiences, thus authors need to explain further how will this contribute to the literature.
Page 3, 2.1. Meta-Ethnography I suggest that authors show the seven-phases methodology for synthesizing qualitative studies using a figure. this is like the PRISMA statement in meta-analysis.
PRISMA diagram should be deleted its indicated for systematic reviews and meta-analyses.
2.5. Overview of synthesized studies for each country provide K=number of papers from a country, n=number of participants e.g. UK K=9, n=.... provide Ref after each country.
Table 2 - add Ref and please add main findings and also a quality score for each paper. quality score is based on critical appraisal. credibility and trustworthiness are key in qualitative work.
Figure 1 is incomprehensible - very poor quality and small text. please fix.
The results section is very detailed.
Discussion please address the following issues:
-Was the recruitment strategy in each study included appropriate to the aims of the research?
-Was the data collected in a way that addressed the research issue?
-Has the relationship between researcher and participants been adequately considered?
How valuable is the meta-ethnography?
The discussion needs to be for and against the researcher’s argument.
Strengths and limitations are adequate.
Conclusion is adequate.
Reviewer 2 Report
- It seems to me that carrying out the review since 2008 is covering an excessive number of years and I would like to know if the reason is that no systematic review has been carried out on the subject and if any systematic review has been carried out, it is important to indicate it.
Reviewer 3 Report
Thank you for giving me the opportunity to review this paper. Overall, the paper is well written, well organized and comprehensively described.
I have a few comments for the authors to consider:
1- table 2: I am leaning toward considering Ethiopia as the country of origin for Aga, Nikkonen & Kylma's work (row 1-3) instead of Finland.
2- Table 2: Row 8: I could not find Monica Barry's paper in the references. I also believe that her study design would fall under qualitative research.
3- Table 2: Row 14: Hagström et al.'s work needs to be revised.
4- Table 2: I strongly believe that the study design column should be revised for consistency and accuracy.
5- the following line is confusing "Studies at the full-text review stage were screened independently by two reviewers against inclusion criteria leaving 28 papers from 25 unique studies."
6- I did not have access to the supplementary file to review. However, I strongly believe that these lines should be revised "wrong study sample (n=30); wrong study design (n=24); wrong experience (e.g., not focused on the experience of being a young carer) (n=22); wrong publication type (n=13); wrong language". Please consider using a term other than "wrong"
